# QTLs and Candidate Genes for Seed Protein Content in Two Recombinant Inbred Line Populations of Soybean

**DOI:** 10.3390/plants12203589

**Published:** 2023-10-16

**Authors:** Hye Rang Park, Jeong Hyun Seo, Beom Kyu Kang, Jun Hoi Kim, Su Vin Heo, Man Soo Choi, Jee Yeon Ko, Choon Song Kim

**Affiliations:** Department of Southern Area Crop Science, National Institute of Crop Science, Rural Development Administration, Miryang 50424, Republic of Korea; hrpark6@korea.kr (H.R.P.); hellobk01@korea.kr (B.K.K.); itomi123@korea.kr (J.H.K.); hsb3937@korea.kr (S.V.H.); mschoi73@korea.kr (M.S.C.); kjeeyeon@korea.kr (J.Y.K.); kcs3925@korea.kr (C.S.K.)

**Keywords:** quantitative trait loci, soybean protein, high protein, genetic map, single-nucleotide polymorphism

## Abstract

This study aimed to discover the quantitative trait loci (QTL) associated with a high seed protein content in soybean and unravel the potential candidate genes. We developed two recombinant inbred line populations: YS and SI, by crossing Saedanbaek (high protein) with YS2035-B-91-1-B-1 (low protein) and Saedanbaek with Ilmi (low protein), respectively, and evaluated the protein content for three consecutive years. Using single-nucleotide polymorphism (SNP)-marker-based linkage maps, four QTLs were located on chromosomes 15, 18, and 20 with high logarithm of odds values (5.9–55.0), contributing 5.5–66.0% phenotypic variance. In all three experimental years, *qPSD20-1* and *qPSD20-2* were stable and identified in overlapping positions in the YS and SI populations, respectively. Additionally, novel QTLs were identified on chromosomes 15 and 18. Considering the allelic sequence variation between parental lines, 28 annotated genes related to soybean seed protein—including starch, lipid, and fatty acid biosynthesis-related genes—were identified within the QTL regions. These genes could potentially affect protein accumulation during seed development, as well as sucrose and oil metabolism. Overall, this study offers insights into the genetic mechanisms underlying a high soybean protein content. The identified potential candidate genes can aid marker-assisted selection for developing soybean lines with an increased protein content.

## 1. Introduction

Soybean [*Glycine max* (L.) Merr.] is an important legume crop globally known for its high-quality protein and oil content [1,2]. Asian countries, such as Korea, Japan, China, and Indonesia, have a strong cultural tradition of consuming soy-based products. Recently, the consumption of traditional soy-based products has surged globally, dominating the global protein market [3,4,5,6]. This substantial growth is attributed to changing dietary preferences and the shifting behavior of consumers towards more sustainable and environmentally friendly food choices [7,8,9,10].

Soybean protein research has gained increasing interest because of its significance. Many researchers have aimed to explore the genetic aspects of the protein traits in soybeans through quantitative trait loci (QTL) and genome-wide association studies (GWAS) [2,4]. The seed protein traits in soybeans are linked with the seed oil content and weight. These quantitative traits are complex and influenced by multiple genes and environmental factors [4,11,12]. In particular, soybean seed storage proteins are influenced by multiple factors, including major transcription factors, phytohormones, protein accumulation, storage protein regulation and deposition, and environmental factors [4]. Since the publication of the soybean reference genome, research on the genetics of these factors has been actively conducted [13]. Researchers have actively reported the genetic regions associated with the protein content using traditional linkage analyses to identify QTLs [11], high-density single-nucleotide polymorphism (SNP) linkage maps for precise QTL detection [4], and GWAS to unravel the genetic basis of soybean protein traits [12,14]. These analyses have been conducted on a wide range of soybean resource populations, genetic resources with a high protein content in backcross, and recombinant inbred line (RIL) populations [12,14,15].

Numerous studies have mapped seed protein and oil content to specific genomic regions, primarily located on chromosomes 15 and 20 [14,16,17,18,19,20]. For example, cqSeed protein-03 has been identified as a major QTL for seed protein on chromosome 20. This QTL has been extensively represented in several large populations using various mapping methods since the publication of the reference genome. However, an accurate identification of the precise location and reliable candidate genes is challenging [11,19,20,21,22,23]. Only recently have some studies successfully fine-mapped cqSeed protein-003 across several mapping populations and narrowed its interval to 77.8 kb [24]. These studies identified an insertion/deletion within the CCT domain of *Glyma.20g085100* and showed a strong correlation with the seed protein content. The function of *Glyma.20g085100* has been confirmed using RNA interference (RNAi) in transgenic soybean plants [24,25].

Furthermore, through the fine mapping of the QTL detected on chromosome 15 [26], a specific allele derived from wild soybean was found to confer simultaneous effects on the 100-seed weight, protein content, and oil content traits that are negatively correlated [19,25,27,28]. This allele was localized to a 329 kb region on chromosome 15 [26]. In addition, *GmSWEET10a*, *GmSWEET10b*, and *GmSWEET39*—other representative genes on chromosome 15—are sugar transporters that affect seed protein and oil content [29,30,31]. *GmST05* (*Glyma.05g244100*) affects both the seed protein and oil content, in addition to its role in controlling seed size. This effect is likely achieved by regulating *GmSWEET39* transcription [32]. However, despite these findings, a comprehensive understanding of the genetic factors influencing soybean protein traits remains elusive.

GWAS signals for protein content were identified on chromosomes 15 and 20, exhibiting a greater prominence in Korean accessions. The frequency of alleles linked to a high protein content was lower in Chinese and US accessions. In Korea, soybean breeding and pedigree programs have focused on breeding for traits specifically related to soy-based food, with a particular emphasis on protein content [33]. Furthermore, the lack of thorough validation for diverse genetic backgrounds and limited utilization in practical breeding programs have been challenging. The main reason for this limitation is the modest effect of these QTLs on phenotypic variation [34]. To overcome this, it is crucial to further validate and evaluate these QTLs for their effective incorporation into breeding programs.

‘Danbaekkong’, previously utilized in several studies, has proven valuable for detecting the QTLs associated with protein content [20,33,35,36]. Several studies have extensively employed ‘Danbaekkong’ in QTL identification and breeding programs utilizing Danbaekkong-derived RIL populations [20,35]. However, Saedanbaek (SD) possesses a genetically distinct background from that of Danbaekkong. The high protein traits found in SD can be traced back to BARC-10 (MD87L, PI 572270), a breeding material recognized for its high protein content that is officially registered in the US National Plant Germplasm System [37]. Therefore, because SD genetically differs from the widely used high-protein cultivar ‘Danbaekkong’, it could unravel new allelic sources to increase the protein content.

The primary focus of this study was to identify the QTLs specifically linked to the seed protein content using two populations of RILs derived from SD, an elite high-protein cultivar as one of the parental lines, over three years. This finding will enhance our understanding of the genetic factors influencing seed protein content and provide valuable insights for future breeding efforts to improve soybean protein traits.

## 2. Results

### 2.1. Phenotypic Variation in the Seed Protein Content

The seed protein contents (%) in the parental lines [YS2035-B-91-1-B-1 (YS2035), Saedanbaek (SD), and Ilmi (IM)] and the two RIL mapping populations [YS2035 × SD (YS) and SD × IM (SI)] were assessed in 2020, 2021, and 2022. The protein contents of YS2035, SD, and IM in the parental lines were 47.3, 54.2, and 42.4%; 46.8, 54.3, and 44.4%; and 43.3, 50.6, and 41.2% in 2020, 2021, and 2022, respectively. The average seed protein content of SD (53.1%) was significantly higher than that of YS2035 (45.6%) and IM (42.7%; Figure 1 and Appendix A). The average protein content in the YS and SI populations ranged from 39.4 to 52.3% and 39.6 to 52.1%, respectively. Substantial variations in the protein content were observed between the parental lines, and the H^2^ values in the YS and SI populations were 0.84 and 0.86, respectively (Appendix A). The protein content (%) in both populations showed a normal distribution and slightly transgressive inheritance. However, this transgressive inheritance was specifically prominent in the SI population, especially in 2021 and 2022 (Figure 1). An analysis of variance (ANOVA) revealed that the year differences and genotype × year interaction effects were highly significant in both the YS and SI populations (Appendix A).

### 2.2. Linkage Map Construction

A total of 180,375 high-quality SNPs markers were genotyped, of which 27,724 in the YS populations and 27,896 in the SI populations were polymorphic between the respective parental lines. After deleting redundant markers with >5% missing values, 2254 and 3544 SNPs were selected and used to construct linkage maps for the YS and SI populations, respectively. The polymorphic SNP markers were distributed across all 20 chromosomes with an average of 113 and 177 markers per chromosome and covered a total of 5339 and 3248 cM genetic distances in the YS and SI linkage maps, respectively. The average distances between the adjacent SNPs in the YS and SI populations were 2.5 and 0.9 cM, respectively. The average lengths (cM) in the YS and SI populations were 267 cM and 162 cM, respectively (Appendix A). The YS population exhibited the lowest number of SNP markers on chromosome 18 (62), while the highest number was found on chromosome 16 (206). In the SI population, the lowest number of SNP markers were observed on chromosome 17 (88), while the highest number were on chromosome 5 (235) (Appendix A). Despite using SD as the common parental line, the differences in the genomic length coverage between the two linkage maps could be attributed to the genetic differences between the other two parental lines (YS2035 and IM). Based on the above results, it was used more accurately for QTL mapping (Appendix A).

### 2.3. QTL Analysis

Across the three years of the experiment, specific QTLs with marker intervals (left–right) for seed protein were detected on chromosomes 15, 16, 17, 18, and 20 in the YS population and on chromosomes 9, 15, and 20 in the SI population (Figure 2 and Appendix A). The differences in detecting different QTLs in the two mapping populations could be attributed to the genetic differences between YS2035 and IM. We selected four QTLs considering the logarithm of odds (LOD) and phenotypic variance explained (PVE) value of five or more years and environment. Subsequently, three out of four QTLs were detected on chromosomes 15, 18, and 20 in the YS population, and one QTL was detected on chromosome 20 in the SI population (Table 1). The LOD of the identified QTLs ranged from 5.9 to 55.0, and the PVE varied from 5.5 to 66.0%. The major QTLs—*qPSD20-1* (LOD, 20.9–30.6; PVE, 22.5–35.4%) in the YS population and *qPSD20-2* (LOD, 23.0–55.0; PVE, 34.1–66.0%) in the SI population on chromosome 20—were consistently detected across all three years. Most of the QTLs identified in relation to the IciM-ADD values were designated as *qPYS16*, as they originated from YS as the parent chromosome 16 in the YS population; however, all of the QTLs were named *qPSD*, because the QTLs appeared in the parent SD regardless of the populations (Appendix A). The major *qPSD20-1* spanned from 31,781,045 to 31,961,695 bp in the YS population. In addition, another major QTL on chromosome 20, *qPSD20-2*, spanning from 30,395,400 to 31,781,045 bp on the physical map, was stably detected for three consecutive years in the SI population. In addition, the QTLs on chromosomes 15 and 18 were detected in more than one year. The physical positions of the markers flanking *qPSD15-1* on chromosome 15 in the YS population detected in 2020 and 2021 were from 7,930,801 to 8,678,412 bp. The physical positions of the QTL *qPSD18-1* detected in 2020 were from 46,911,930 to 47,526,734 bp. A total of 181 genes were identified in the four QTL regions (Table 1 and Appendix A).

### 2.4. Phenotypic Variation According to the Allele Patterns

The top 20 and bottom 20 RILs with high and low seed protein content were selected from the YS and SI populations (Table 2). These genotypes were assessed using representative markers linked to *qPSD15-1*, *qPSD18-1*, and *qPSD20-1*, located on chromosomes 15, 18, and 20, respectively, which are associated with genes that promote a high protein content.

The average protein content in the RILs with SD genotypes (high protein) was 53.1%, while that in the RILs with the YS and IM genotypes (low protein) was 45.6% and 42.7%, respectively, over the three years (Appendix A). The protein content in the top 20 RILs ranged from 51.1 to 52.3%, whereas that in the bottom 20 RILs ranged from 39.4 to 41.6% in both the YS and SI populations (Appendix A). Through the genome sequencing of the three parents—SD, YS, and IM—both populations included SD, and the QTL regions were all derived from SD, so the RILs in the SI population were included in the top 20 proteins. An analysis of the allelic patterns in the top 20 RILs revealed that the SD was predominantly present in these recombinants at the three loci. In contrast, the alleles of the bottom 20 recombinants at the representative markers were mostly derived from IM or YS (Table 2). In both populations, the RILs with the SD allele in the *qPSD15-1* marker exhibited an average protein content of 46.6%, whereas those with the YS or IM allele showed a protein content of 45.2% (Figure 3a). Similarly, the RILs with the SD allele for the *qPSD18-1* marker had an average protein content of 47.0%, whereas those with the YS or IM allele displayed 45.5% (Figure 3b). The RILs with the SD allele at the *qPSD20-1* marker had an average protein content of 49.3%, whereas those with the YS or IM allele had a protein content of 44.9% (Figure 3c). According to the combination of the allele patterns of *qPSD15-1*, *qPSD18-1*, and *qPSD20-1*, the protein content of the RILs with all low-protein alleles (AAA) was 43.3%, whereas that of the RILs with all high-protein parental alleles (BBB) was 49.6%. The RILs with an SD allele at *qPSD15-1* (BAA) exhibited an average protein content of 44.6%, while those with an SD allele at *qPSD18-1* (ABA) had a protein content of 44.0%. The RILs with an SD allele at *qPSD20-1* (AAB) showed a protein content of 48.1%. Furthermore, recombinants harboring SD alleles at *qPSD15-1* and *qPSD18-1* (BBA) had an average protein content of 45.7%. Similarly, the protein content in the RIL with SD alleles at both *qPSD18-1* and *qPSD20-1* (ABB) was 48.9%, while that in the RILs with SD alleles at both *qPSD15-1* and *qPSD20-1* (BAB) was 48.6% (Figure 3d).

### 2.5. SNP Variation Analysis and Variant Annotation

Next, we annotated the polymorphic SNPs in the genes mapped to the *qPSD15-1*, *qPSD18-1*, and *qPSD20-1* QTLs using the whole-genome sequencing of SD, YS2035, and IM. After verifying the tri-parent SNP selection based on the soybean reference genome, only genes that exhibited differences from SD were screened for SNPs in YS2035 and IM. In total, 28 genes were selected among the 181 genes mapped to the intervals of the four QTLs (Table 3). The variant annotations identified 9 frameshift, 96 missense, 4 stop-gains, and other variants in 89 genes mapped to *qPSD15-1*. The genes mapped to *qPSD18-1* (39) comprised 6 frameshift, 91 missense, 3 stop-gains, and others. In all three experimental years, *qPSD20-1* and *qPSD20-2* were consistently identified in the overlapping genomic regions in the YS and SI populations. *Glyma.20g085100* was selected in a common SNP from two populations as the missense variant. In the *qPSD20-1* and *qPSD20-2* regions, we identified 7 and 46 genes, respectively. In these genes, we detected 1 frameshift, 55 missense, 1 stop-gain, and more (Table 1 and Appendix A). Of the 28 annotated genes in the QTL regions, 6 harbored stop-gain, 13 harbored frameshift, and 9 harbored missense variants (Table 3). The annotation of these 28 genes identified their association with several biological processes, including starch biosynthetic, carbohydrate metabolic process, sucrose metabolic process, fatty acid biosynthesis, lipid metabolic process, and protein polymerization (Table 3).

## 3. Discussion

This study aimed to discover new genes associated with the protein content in soybeans using two RIL populations derived from soybean cultivars with contrasting protein contents. Among the three parental lines, SD—developed in 2010 as a high-protein cultivar (48.2%) in South Korea—is widely recommended for soybean foods, such as tofu and soybean paste [44]. Here, the average protein content measured in SD over three years was 53.1% (Appendix A), whereas the average protein content of cultivated soybeans is approximately 40% [45]. A high broad-sense heritability for protein content was observed in average years 0.84 and 0.86 in the YS and SI populations, respectively (Appendix A), suggesting a highly significant (*p* < 0.001) influence of genotype × year interaction on the traits. These results suggest that SD is a suitable candidate for conducting QTL analyses to map the genetic intervals associated with a high protein content in soybeans.

Since the first study on the QTLs associated with protein content—which identified cqProt-001 and cpProt-003 on chromosomes 15 and 20, respectively [16]—several studies have confirmed the involvement of these QTLs in regulating the protein content in soybeans [33]. Additionally, other QTLs have also been identified in soybeans. A QTL related to a high protein and low oil content contributed by PI407788A, a high protein cultivar, was identified on chromosome 15 [17]. The QTL, cqSeed protein-003, located on chromosome 20, is associated with protein and amino acid content and derived from another high-protein cultivar, Danbaekkong [20,35]. Bandillo et al. [46] used SoySNP50K data to explore the connection between genetic variations and protein content across more than 12,000 *G. max* accessions [47].

This study detected a high LOD value, PVE, and stability of the major QTLs *qPSD20-1* in the YS population and *qPSD20-2* in the SI population. The major seed protein content QTLs on chromosome 20, commonly referred to as the repeat overlapping interval, have been identified in numerous studies [14,15,19,24,25,41]. In other RIL populations derived using SD as a parental line, *qHPO20*—associated with seed protein and oil content, and mapped to a wide region (4.8–34.3 Mbp) on chromosome 20—was stably detected in three years [19]. Our study located *qPSD20-1* and *qPSD20-2* to narrower intervals (31.7–31.9 and 30.3–31.7 Mbp, respectively; Table 1) than those in previously reported studies. Concordant with our study, previous studies have identified major protein- and oil content-related QTLs and confirmed the association of genes with the traits on chromosome 20 [11,12,14,15,17,19,20,24,41]. Our stable and major QTLs on chromosome 20 identified here harbored eight genes in the YS and SI populations. In particular, *Glyma.20g085100* is an SNP found commonly in both populations. Another study identified *Glyma.20g085100*, underlying the major QTL located on chromosome 20, related to soybean seed protein and oil, harboring tandem repeats. This gene encodes the CCT domain [14,24,25]. The CCT-domain gene, *POWR1*, likely related to lipid metabolism and nutrient transport, plays a pleiotropic role in regulating soybean seed quality and yield [25]. The insertion of a transposable element into the CCT domain of *POWR1* led to an increased seed weight and oil content but decreased protein content. Conversely, the overexpression of *POWR1* in transgenic plants improved protein content but reduced seed weight and oil content [25]. Among these, one gene exhibited a stop-gain, and another showed a missense variant in the YS population, whereas seven genes displayed a missense variant in the SI population. (Table 3). *Glyma.20g081500* (lipase-containing protein) and *Glyma. 20g082700* (sugar efflux transporter SWEET52) are presumed to be involved in protein, carbohydrate, and lipid metabolism during soybean seed development. These studies have shown that these genes would affect protein content after seed maturity [38,42,43,48,49]. However, there are few specifically studied and identified genes within this interval. These genes have not been characterized in previous studies; therefore, understanding their role in regulating soybean protein content warrants further research.

The QTL *qPSD18-1* on chromosome 18 in the YS population was detected at 46.9–47.5 Mbp intervals in 2022 (Table 1 and Appendix A). Among the genes underlying these QTLs, two displayed stop-gains, five showed frameshift variants, and three exhibited missense variants. *Glyma.18g193600* (fructose-1,6-bishosphatase) is thought to be related to seed sucrose development (Table 3). A recent GWAS study reported that *Glyma.18g193600* is likely to play a role in the interconnected process of sucrose biosynthesis in edamame beans [38]. Among the potential candidate genes identified here, *Glyma.18g195700, Glyma.18g195900,* and *Glyma.18g196000* (fatty acid biosynthesis) might be related to soybean storage proteins [39,40]. In soybean seeds, storage proteins— essential nutritional components—are initially synthesized as precursors in sucrose and oil [38,39,49].

The QTL *qPSD15-1* on chromosome 15 in the YS population, as a novel QTL, was detected at 7.9–8.6 Mbp intervals in 2020 and 2021 (Table 1 and Appendix A). Among this QTL, three displayed stop-gains and seven exhibited frameshift variants. *Glyma.15g108000* (the starch/carbohydrate-binding module) and *Glyma.15g108900* (carbohydrate metabolic process) are involved in carbohydrate biosynthesis, and related genes are being published in chromosome 15 (Table 3). Recently, *Glyma.15g049200* was identified as one of the candidate genes through fine mapping within the QTL regions simultaneously associated with soybean seed weight, protein content, and oil content [26,31]. Moreover, the QTLs on chromosome 15 exhibit pleiotropic effects on soybean seed protein and oil content. Certain sugar transporters, such as *GmSWEET10a*, *GmSWEET39* (*Glyma.15g049200*), and *GmSWEET10b* (*Glyma.8g183500*) have been identified in these regions [31,35]. During soybean domestication, the SWEET paralogs *GmSWEET10a* and *GmSWEET10b* went through stepwise selection, influencing seed size, oil, and protein levels by regulating the sugar distribution from the seed coat to the embryo [31,33]. In addition to the major QTL, the minor QTLs on chromosome 15 with overlapping positions, as detected in previous studies [50], may also contribute to seed protein and oil content.

During soybean seed development, storage proteins are transported for carbohydrate and lipid synthesis [4,32]. The genes *Glyma.15g108000*, *Glyma.15g108900, Gylma.18g193600*, and *Glyma.20g082700* (related to starch and carbohydrates synthesis during seed development), *Glyma.18g195700, Glyma.18g195900*, *Glyma.18g196000* (associated with fatty acid biosynthesis), and *Glyma.20g081500* (related to lipid catabolic processes during seed development) are likely to regulate protein accumulation. These candidate genes may regulate protein accumulation by influencing the sugar delivery from the seed coat integument to the embryo [38,40,43]. Several studies on soybean RIL populations have reported that seed protein, sucrose, and oil content show negative correlations [4,19,25,27,28,32,51]. Most candidate genes identified in this study have not been previously reported to be associated with soy protein. Therefore, further studies are required to gain valuable insights for soybean protein research.

QTLs related to seed protein content have been extensively studied using GWAS, QTL analyses, fine mapping, and haplotype mapping [11,12,14,15,19,20,24,25,46]. Here, we identified novel regions on chromosomes 15, 18, and 20, which showed consistent associations with soybean protein contents. These findings suggest that the newly identified QTLs, along with previously recognized ones, are likely to further elucidate the genetic factors associated with protein-related traits. However, it remains difficult to identify genes that are directly involved in regulating protein traits, warranting further studies using genetic resources with a high protein content.

## 4. Materials and Methods

### 4.1. Plant Materials

Two RIL populations involving three parental lines: SD (high-seed protein cultivar) [44], YS2035 (low-seed protein line), and IM (low-seed protein cultivar) [52], were used here. The YS (YS2035 × SD) and SI (SD × IM) mapping populations were developed using the single-seed descent method from the F2 to the F5:10 and F5:7 generations, respectively. The YS and SD mapping populations, comprising 237 and 189 RILs, respectively, and the parental lines were cultivated under experimental field conditions at the Miryang farm in South Korea (35°29′46.5″ N 128°44′29.9″ E) in 2020, 2021, and 2022. The populations were planted in rows measuring 4 m in length, with spacings of 70 cm between each row and 15 cm between individual plants. Fertilizers and pesticides were administered following established cultivation methods in South Korea [53].

### 4.2. Analysis of Crude Seed Protein Concentrations

The protein content was measured using 15 mg of seed powder in both mapping populations, comprising 426 RILs, and assessed each year using the Dumas method [54] with a Rapid N Cube (Elementar Analysen System, Hanau, Germany), following the manufacturer’s instructions [55]. The protein analysis of each individual RIL was performed three times per year.

### 4.3. Genomic DNA Extraction and Genotyping

Genomic DNA was extracted from dry seeds of each line of the mapping populations and the three parental lines using a Maxwell RSC 48 instrument (Promega Madison, WI, USA), following the manufacturer’s instructions. The DNA quality was assessed using a NanoDrop ND-2000 (Thermo Fisher Scientific, Waltham, MA, USA), and each DNA sample was diluted to a concentration of 10 ng/μL for genotyping. The mapping populations and the parental lines were genotyped using the 180K Axiom SoyaSNP array [56].

### 4.4. Genetic Linkage Map Construction and QTL Analysis

The SNP markers showing polymorphism between the parental lines were identified from the Axiom 180 K SoyaSNP array genotyping data to construct the genetic linkage map. The genetic linkage maps of the two mapping populations were constructed using the QTL IciMapping software version 4.2 [57]. The grouping threshold was set at a 3.0 logarithm of odds (LOD), nnTwoOpt was used as the ordering algorithm, and the sum of the adjacent recombination fractions was used for rippling, following the methodology described in a previous study [37]. Missing data with >5% were used to remove the redundant markers. The mapping of each linkage group was performed using Kosambi’s mapping function. The association between each trait and the SNP markers was assessed using the inclusive composite interval mapping (IciM) function of the IciMapping software, with a 1000 permutation test. The QTLs were named by combining abbreviated letters *q* for QTL, *P* for seed protein, and *SD* for the parent Saedanbaek (SD), followed by the chromosome name and nth QTL on the chromosome. For instance, *qPSD*15-2 represents the second QTL identified on chromosome 15.

### 4.5. Prediction of Novel Candidate QTL and Genes

Firstly, the QTLs detected for more than two years were selected. Statistically significant QTLs associated with soybean seed protein content were identified by examining the genotypes within the QTL regions using SNP markers. We performed the genome sequencing of SD, YS2035, and IM using the Illumina Hiseq X sequencing platform (Illumina, San Diego, CA, USA). Reads were mapped using Bowtie 2 (v2.2.4) and variants were called with Freebayes (v1.3.4). After verifying tri-parent SNP selection based on the soybean reference genome, only genes that exhibited differences from SD were screened for SNPs in YS2035 and IM. The QTL regions were further investigated using SoyBase (www.soybase.org (accessed on 5 September 2023)) to identify the candidate genes. Annotated information on the candidate genes was obtained from the soybean reference genome (Wm82. a4. v1). The candidate genes were presented based on their gene descriptions and SNP variations within the QTL regions.

### 4.6. Statistical Analysis

To assess the phenotypic variations in protein within the populations, various statistical tests were performed, including an analysis of variance (ANOVA), Student’s *t*-test, and Duncan’s multiple range test (DMRT). The statistical analyses were conducted using R V3.6.3 software [58]. The broad-sense heritability (H^2^) for the mean values in each environment was calculated using an equation with some modifications [59].
H^2^ = *σ*^2^*_G_*/(*σ*^2^*_G_*+ *σ*^2^*_GY_*/*Y* + *σ*^2^*_e_*_/*rY*_)(1)
where *σ*^2^_*GY*_ and *σ*^2^*_e_* are the components of genotype × year and error variances, respectively. The component of genotype × year variance (*σ*^2^*_GY_*) and the mean square of error (*σ*^2^*_e_*) was estimated with reference [60].

## 5. Conclusions

We conducted a three-year field study using two RIL populations derived from a cross between the elite cultivar SD and either YS2035 or IM and identified several QTLs on chromosomes 15, 18, and 20. In all three experimental years, *qPSD20-1* and *qPSD20-2* were consistently identified in the overlapping genomic region in the YS and SI populations. These QTLs have been previously reported in various studies related to soybean protein content, whereas the other identified QTLs are novel. This suggests that the regulation of protein content in soybean seed may be influenced by sucrose and oil biosynthesis. Therefore, the potential utility of the results from this study for protein in soybean seed is expected to increase.

## Figures and Tables

**Figure 1 plants-12-03589-f001:**
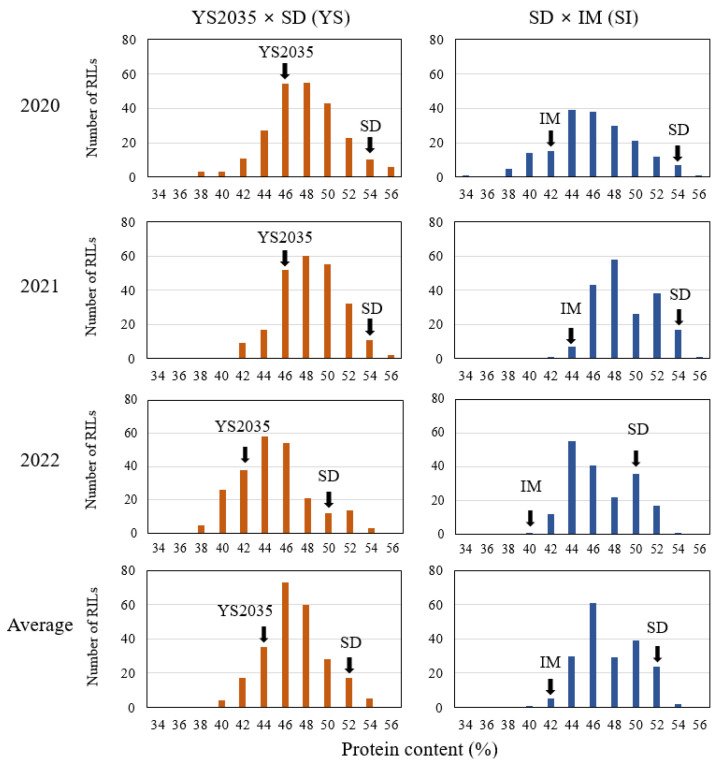
Frequency distribution of RIL protein content in the two mapping populations evaluated in 2020, 2021, and 2022. The parental values are shown using arrows. YS2035; YS2035-B-91-1-B-1, SD; Saedanbaek, IM; Ilmi, YS; YS2035 × SD, SI; and SD × IM.

**Figure 2 plants-12-03589-f002:**
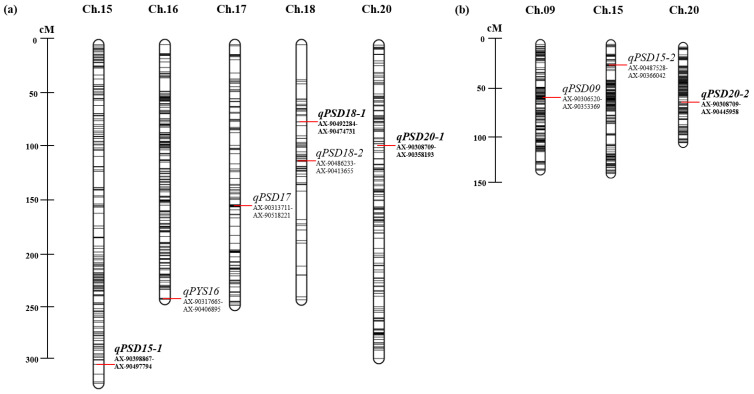
Quantitative trait loci (QTL) associated with seed protein content in (**a**) YS2035 × Saedanbaek (YS) and (**b**) Saedanbaek × Ilmi (SI) mapping populations. The bars inside each chromosome represent the position of markers used to construct the linkage map. The QTLs and marker positions are shown using red bars. The genetic distance (cM) of chromosomes was displayed as the rulers on the left side in the YS and SI populations. The main selected QTLs are highlighted by the bold font. Genetic map details are provided in Appendix A.

**Figure 3 plants-12-03589-f003:**
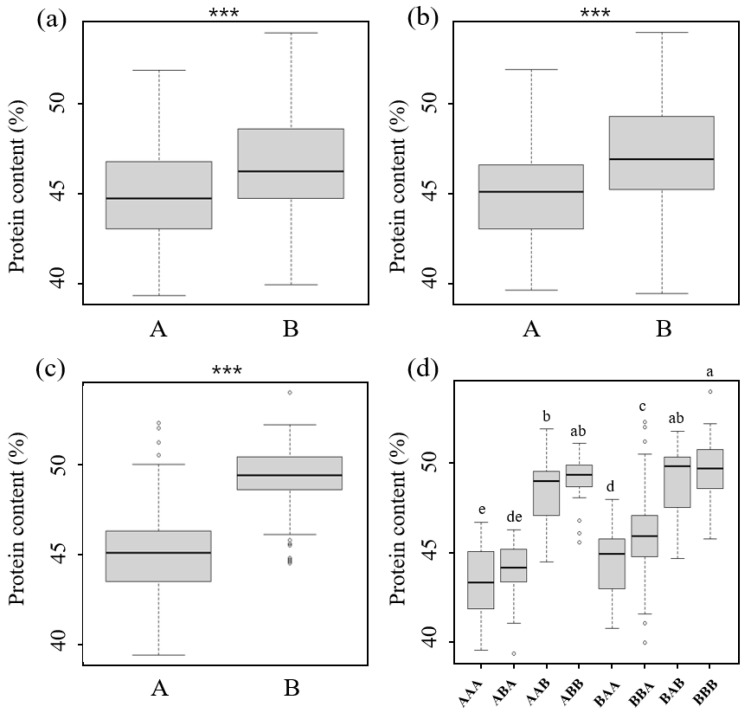
Boxplots of protein content and the allele effect of the major QTLs on chromosomes 15, 18, and 20. Trait values of the recombinants with high (SD; B) or low parent (YS/IM; A) alleles (**a**) *qPSD15-1*, (**b**) *qPSD18-1*, (**c**) *qPSD20-1* markers and (**d**) with all three markers. Asterisks indicate significant differences between parental lines in the RILs of ‘YS2035’ and ‘Saedanbaek’ (YS) or ‘Saedanbaek’ and ‘Ilmi’ (SI) at *p* < 0.001. The center bold line represents the median. Different lowercase letters indicate significant differences between genotypes; *p* < 0.05; Duncan’s multiple range test (DMRT).

**Table 1 plants-12-03589-t001:** Quantitative trait loci (QTL) associated with high protein identified in the two recombinant inbred line (RIL) mapping populations derived from ‘YS2035’ × ‘Saedanbaek’ and ‘Saedanbaek’ × ‘Ilmi’.

Population ^1^	Marker ^2^	Chr ^3^	Genetic Position (cM)	Physical Position of Markers (bp) ^4^	Year	Gene Name	Gene No.	LOD ^5^	PVE ^6^ (%)	Add ^7^	Reference
Y × S	*qPS* *D15* *-1*	15	305	7,930,801–8,678,412	2020 2021 Average ^8^	*Glyma.15g101800–Glyma.15g110600*	89	14.0 14.6 12.3	17.5 17.1 13.8	−2.7 −2.1 −1.7	
Y × S	*qPSD18* *-1*	18	75	46,911,930–47,526,734	2022 Average	*Glyma.18g19* *3300–Glyma.18g* *197100*	39	6.7 5.9	7.0 5.5	−0.9 −0.6	[38,39,40]
Y × S	*qPSD20-1*	20	96	31,781,045–31,961,695	2020 2021 2022 Average	*Glyma.20g085100–Glyma.20g085700*	7	21.1 24.7 20.9 30.6	22.5 29.1 24.7 35.4	−1.8 −1.6 −1.8 −1.6	[14,24,25]
S × I	*qPSD20-2*	20	68	30,395,400–31,781,045	2020 2021 2022 Average	*Glyma.20g08* *1000–Glyma.20g0* *85450*	46	23.0 48.2 55.0 52.7	34.1 59.7 66.0 61.5	2.4 2.1 2.4 2.3	[14,15,19,24,25,41]

^1^ Y × S, YS2035 × Saedanbaek; S × I, Saedanbaek × Ilmi. ^2^ *qPSD*, ‘Saedanbaek’ contributed to the allele. ^3^ Chr, Chromosome. ^4^ Physical position of the markers, the soybean reference genome (*Glycine max* Wm82.a2.v1) was used to determine the physical positions of the markers. ^5^ Logarithm of odds value at the peak likelihood of QTL. ^6^ Phenotypic variation explained (PVE) by QTL. ^7^ Additive effect. ^8^ Average values for three years: 2020, 2021, and 2022.

**Table 2 plants-12-03589-t002:** Genotypes of the top and bottom 20 RILs in YS2035 × Saedanbaek (YS) and Saedanbaek × Ilmi (SI) populations based on high and low protein content at markers linked to *qPSD15-1*, *qPSD18-1*, and *qPSD20-1* markers. ‘A’ genotype shows that the selected RILs derived line was homogeneous for the allele from YS2035-B-91-1-B-1 (YS: low protein) and Ilmi (IM: low protein), ‘B’ genotype shows that the line was homogeneous for the allele from Saedanbaek (SD: high protein).

Top 20 RILs with High Protein Content	Genotype of the Marker Linked to the QTLs			Protein Content (%)	Bottom 20 RILs with Low Protein Content	Genotype of the Marker Linked to the QTLs			Protein Content (%)
	qPSD15-1	qPSD18-1	qPSD20-1			qPSD15-1	qPSD18-1	qPSD20-1	
YS-196	B	B	A	52.3	YS-229	A	B	A	39.4
SI-400	B	B	B	52.1	SI-465	A	A	A	39.6
YS-068	B	B	B	52.1	YS-111	A	A	A	39.8
YS-080	B	A	B	52.0	YS-209	A	A	A	39.9
YS-005	B	A	B	52.0	YS-036	B	B	A	40.0
SI-317	A	A	B	51.9	YS-037	A	A	A	40.3
YS-190	B	A	B	51.8	YS-117	A	A	A	40.4
YS-109	B	B	B	51.7	SI-337	A	A	A	40.6
YS-199	B	B	B	51.6	YS-043	B	A	A	40.8
SI-428	B	B	B	51.5	YS-118	B	A	A	40.9
YS-173	B	A	B	51.4	SI-361	B	B	A	41.1
SI-423	B	A	B	51.4	YS-063	A	B	A	41.1
YS-205	B	A	B	51.3	YS-227	B	A	A	41.2
SI-445	B	B	B	51.3	YS-048	B	A	A	41.2
YS-090	B	B	B	51.3	SI-473	B	A	A	41.2
YS-008	B	A	B	51.2	YS-015	B	A	A	41.4
SI-348	B	B	B	51.2	YS-235	A	A	A	41.6
YS-202	B	B	B	51.2	YS-100	B	B	A	41.6
SI-326	B	B	B	51.1	SI-502	A	A	A	41.6
SI-345	A	B	B	51.1	YS-045	B	A	A	41.6

**Table 3 plants-12-03589-t003:** Candidate genes for seed protein content identified on the reference genome based on the QTL-linked SNPs in the YS and SI mapping populations.

Population	Marker	Gene ID	Annotation Description	Biological Process	Reference	SNP Type
Y × S	*qPSD15-1*	*Glyma.15g102100*	Alpha/Beta hydrolase domain-containing protein	NA		Stop gain
		*Glyma.15g102202*	Elongation factor Tu GTP binding domain	Translational elongation		Frameshift variant
		*Glyma.15g102252*	Elongation factor Tu C-terminal domain	Translational elongation		Frameshift variant
		*Glyma.15g102800*	Mediator of RNA polymerase II transcription subunit 33a	Phenylpropanoid metabolic process		Stop gain
		*Glyma.15g103100*	Mitochondrial editing factor 18	RNA modification		Frameshift variant
		*Glyma.15g107200*	GPI-anchored protein	Biological process		Stop gain
		*Glyma.15g108000*	Starch/carbohydrate-binding module (family 53)	Starch biosynthetic process		Frameshift variant
		*Glyma.15g108900*	Glycosyl hydrolases family 17	Carbohydrate metabolic process		Frameshift variant
		*Glyma.15g109800*	Peroxisomal membrane protein 2	Biological process		Frameshift variant
		*Glyma.15g109900*	F-BOX protein with a domain protein	NA		Frameshift variant
	*qPSD18-1*	*Glyma.18g193300*	Laccase	Iron ion transport		Frameshift variant
		*Glyma.18g193600*	Fructose-1,6-bisphosphatase, N-terminal domain	Sucrose metabolic process	[38]	Frameshift variant
		*Glyma.18g194700*	NA	NA		Stop gain
		*Glyma.18g194900*	NA	NA		Frameshift variant
		*Glyma.18g195000*	NA	Biological process		Frameshift variant
		*Glyma.18g195700*	Alpha-carboxyltransferase aCT-1 precursor	Fatty acid biosynthesis	[39,40]	Missense variant
		*Glyma.18g195900*	Carboxyl transferase domain	Fatty acid biosynthesis	[39,40]	Missense variant
		*Glyma.18g196000*	Carboxyl transferase domain	Fatty acid biosynthesis	[39,40]	Missense variant
		*Glyma.18g196600*	NA	NA		Stop gain
		*Glyma.18g197100*	NA	NA		Frameshift variant
	*qPSD20-1*	*Glyma.20g085100*	*POWR1* CCT motif family protein	Biological process	[14,24,25]	Missense variant
		*Glyma.20g085700*	Unknown protein	NA	[15]	Stop gain
S × I	*qPSD20-2*	*Glyma.20g081500*	Lipase containing protein	Lipid catabolic process		Missense variant
		*Glyma.20g082450*	Ammonium transporter 1	Ammonium transport	[15]	Missense variant
		*Glyma.20g082700*	Sugar efflux transporter SWEET52	Carbohydrate transport	[42,43]	Missense variant
		*Glyma.20g084000*	Small nuclear ribonucleoprotein F	Spliceosomal snRNP assembly	[15]	Missense variant
		*Glyma.20g084051*	Far1-relate	Regulation of transcription	[15]	Missense variant
		*Glyma.20G084500*	WD40 repeat protein	Innate immune response	[15]	Missense variant
		*Glyma.20g085100*	*POWR1* CCT motif family protein	Biological process	[14,24,25]	Missense variant

The soybean reference genome (*Glycine max* Wm82.a4.v1) was used to annotate genes.

## Data Availability

The data sets generated in this study are included in this published article and its Appendix A.

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
