# Peer review of "QTLs and Candidate Genes for Seed Protein Content in Two Recombinant Inbred Line Populations of Soybean"

_plants, 2023, doi:10.3390/plants12203589_

Round 1

Reviewer 1 Report (New Reviewer)

1. No consensus QTL was identified in the two populations, why are the four QTLs listed in 168-170 mentioned to be repeatedly identified in both of the two populations?

2. A total of 9 QTLs were identified in the two populations, and 4 loci were selected for detailed analysis. What was the basis for the selection of 4 loci? If the selected loci were identified in multiple environments (including Average), why not add qPSD09 detected in 2022 and average?

3. qPSD15-1, qPSD18-1, and qPSD20-1 were identified in Y×S population, is it appropriate to select materials in SI population when analyzing the effects of the three loci combinations?

4. Suggest labeling a ruler in Figure 2.

No

Author Response

  1. No consensus QTL was identified in the two populations, why are the four QTLs listed in 168-170 mentioned to be repeatedly identified in both of the two populations?

We appreciate your review and question. We misdirected the four QTLs shown in Table 1 as they were taken from the two populations. However, this is the sentence that needs to be modified, and since further explanation is needed, we will revise it by adding the following sentence on.

->line 170

Subsequently, three out of four QTLs were detected on chromosomes 15, 18, and 20 in the YS population, and one QTL was detected on chromosome 20 in the SI population (Table 1).

  1. A total of 9 QTLs were identified in the two populations, and 4 loci were selected for detailed analysis. What was the basis for the selection of 4 loci? If the selected loci were identified in multiple environments (including Average), why not add qPSD09 detected in 2022 and average?

Referring to Supplemental Table 5, 9 QTLs were detected in two populations. Among them, the main QTLs were selected as mentioned below. Therefore, the following sentence was added on.

->line 167

We selected four QTLs considering the logarithm of odds (LOD), and phenotypic variance explained (PVE) values of five or more, and year environment. Subsequently, three out of four QTLs were detected on chromosome 15, 18, and 20 in the YS population, and one QTL was detected on chromosome 20 in the SI population (Table 1).

qPSD09 was detected in 2022 and averaged; however, its LOD and PVE values were 6.2 and 5.5, and 3.8 and 3.5, respectively. Its PVE value was not selected because it did not exceed the standard of 5. So, we did not include qPSD09.

  1. qPSD15-1qPSD18-1, and qPSD20-1were identified in Y×S population, is it appropriate to select materials in SI population when analyzing the effects of the three loci combinations?

In the SI population, qPSD20-2 was overlapped because a section similar to qPSD 20-1 of the YS population appeared. In addition, we were able to detect the genome type in SNP markers from each through genome sequencing of parents SD, YS, and IM. Moreover, the two populations contained SD at the same time, and in particular, QTLs derived from SD were identified in eight QTLs, excluding chromosome 16.

Therefore, for this reason, we examined the genomic types of SNPs in the top 20 groups of protein per group in qPSD15-1 and qPSD18-1 by mixing Y×S and S×I populations. Therefore, the following sentence was added on.

->line 204

Through genome sequencing of the three parents SD, YS, and IM—both populations included SD, and the QTL regions were all derived from SD, so RILs in the SI population were included in the top 20 proteins.

  1. Suggest labeling a ruler in Figure 2.

We marked the left side of each group with a Ruler to compare chromosome sizes as per your recommendation. A description was added to Figure 2 (on line 236) as follows.

->line 236

The rulers on the left side display the genetic distance (cM) of chromosomes in the YS and SI populations.

Comments on the Quality of English Language

"The main text is clear, with the occasional spelling or grammatical error mostly confined to the Materials sections and within the Figures which are not pointed out individually here but need to be corrected."

These spelling and grammatical errors have been checked and corrected in the entire document.

Reviewer 2 Report (New Reviewer)

In line 394, the authors mentioned ICIM was used. Could you explain why this method was used? I know there are many QTL mapping methods. Do you think if it is better to add more methods in the analysis to make the results more reliable?

In line 422, the equation should be modified.

Minor editing of English language required.

Author Response

Reviewer2

In line 394, the authors mentioned ICIM was used. Could you explain why this method was used? I know there are many QTL mapping methods. Do you think if it is better to add more methods in the analysis to make the results more reliable?

The Inclusive Composite Interval Mapping (IciM) method is the most advanced and universally adopted technique for conducting QTL analysis in the field of agriculture. It has now become available under a free license, making it widely accessible. IciM surpasses other methods, such as Simple Interval Mapping, by offering the ability to pinpoint QTLs with greater precision and effectiveness, enabling researchers to analyze specific genomic regions more comprehensively. So, we analyzed QTL for the seed protein content of two populations using the IciM method. Even if many related references were searched, the analysis method was used without doubt using IciM software version 4.2 with the IciM-ADD for building high-density linkage maps and mapping QTL in biparental populations.

In line 422, the equation should be modified.

We modified the broad-sense heritability value from 84 and 86 to 0.84 and 0.86 in the result, discussion, methods and added the following on

->line 438

The broad-sense heritability (H2) for mean values in each environment was calculated using an equation with some modifications [59].

H22G/(σ2G + σ2GY/Y + σ2e/rY), (1)

where σ2GY and σ2 e are the components of genotype × year and error variances, respectively. The component of genotype × year variance (σ2GY) and the mean square of error (σ2e) was estimated with reference [60].

Comments on the Quality of English Language

Minor editing of English language required.

The language has been edited thoroughly for grammatical, and other errors.

Round 2

Reviewer 1 Report (New Reviewer)

No

Author Response

The manuscript has improved tremendously after review; however, I am asking the corresponding author to please:

1. Provide the genetic linkage maps for both RIL populations in the Supplementary data and refer to them throughout in the manuscript text. Figure 2 is good, but oversimplified.

We appreciate your review and question. We added the high-density genetic linkage map as supplementary table S5. And we also revised it by adding the following sentences in result and figure 2.

->line 168 and 245

2. Provide more details describing the genetic linkage maps in Results section 2.2. Linkage Map Construction.

We added more details about linkage map and revised them, and added the following sentences.

->line 158

Thank you.

This manuscript is a resubmission of an earlier submission. The following is a list of the peer review reports and author responses from that submission.

Round 1

Reviewer 1 Report

This manuscript provides some valuable findings. However, the presentation and clarity of the results and data in this paper are obviously insufficient, which needs to be carefully sorted out and corrected by the author.

1For example: Table 2 does not show the important data described in the text, and I do not know why tables 1 and 3 are placed behind? For the same parental SD, the obvious difference between the two sets of data for protein content also needs to be explained in detail. Similar problems may exist in other charts, and the chart should be as self-evident as possible.

2It is suggested that the genetic effects of related candidate genes should be analyzed and annotated to improve the reference value of the paper.

3If possible, it is recommended to supplement the results of functional identification experiments for key candidate genes.

Author Response

Special Issue Editor

Plants

Special Issue: QTL Mapping of Seed Quality Traits in Crops

Dear Editor:

We wish to re-submit the manuscript titled “QTLs and Candidate Genes for Seed Protein Content in Two Recombinant Inbred Line Populations of Soybean.” The manuscript ID is plants-2522650.

We thank you and the reviewers for your thoughtful suggestions and insights. The manuscript has benefited from these insightful suggestions. I look forward to working with you and the reviewers to move this manuscript closer to publication in the Plants; Special Issue: QTL Mapping of Seed Quality Traits in Crops.

The manuscript has been rechecked, and the necessary changes have been made following the reviewers’ suggestions. The responses to all comments have been prepared and attached herewith/given below. We hope the revisions and responses have satisfactorily addressed the concerns raised by the reviewers.

I appreciate your consideration. I look forward to hearing from you.

Sincerely,

Jeong-Hyun Seo

Upland Crop Breeding Research Division, Department of Southern Area Crop Science, National Institute of Crop Science, Rural Development Administration

Miryang, Republic of Korea, 50424

+82-55-350-1277

+82-55-353-3050

[email protected]

Response to Reviewer1 comments

This manuscript provides some valuable findings. However, the presentation and clarity of the results and data in this paper are obviously insufficient, which needs to be carefully sorted out and corrected by the author.

Response: Dear Reviewer, thank you for providing valuable feedback on our work. We sincerely appreciate your comments and the time and consideration you dedicated to reviewing our paper. We have made the necessary changes following them and are eager to receive your valuable suggestions, as your expertise is highly valued. Your thoughtful feedback will be of great help to us.

  1. For example: Table 2 does not show the important data described in the text, and I do not know why tables 1 and 3 are placed behind? For the same parental SD, the obvious difference between the two sets of data for protein content also needs to be explained in detail. Similar problems may exist in other charts, and the chart should be as self-evident as possible.

Response: We believe that conducting an ANOVA analysis between the two groups presented in Table 2 is essential for our QTL research. Additionally, numerous studies highlight the importance of evaluating the group sizes and their respective measures. Therefore, obtaining this data is deemed necessary for our research. We kindly request the consideration of these points from the reviewers to enhance the robustness of our study. Nevertheless, to ensure clarity, we revised the related sentences as follows:

(Line113) add

“As shown in Table 1, analysis of variance (ANOVA) revealed that the year differences and the genotype × year interaction effects were highly significant in both YS and SI populations (p < 0.001). Across all years, the mapping population displayed a higher average protein content.”

-> why tables 1 and 3 are placed behind?

Response: Dear reviewer, thank you for bringing this to our attention. We apologize for the oversight on our part. The alignment of the Tables may have been disrupted during the upload process, even though they appeared fine before submission. We sincerely regret any inconvenience this might have caused. We have revised the manuscript and ensured all Figures and Tables are correctly placed.

-> Table 1 protein contents in two populations

Response: It is interesting to observe that the protein content percentages of the new protein vary slightly each year, with the results from 2022 being notably lower. However, it is crucial to consider that protein content is influenced by environmental factors, which is why analyzing the trend over three years of cultivation data is important.

We acknowledge the concerns you raised; indeed, the protein content can be influenced by various factors. Therefore, we conducted a three-year cultivation to demonstrate the trends in protein content.

However, it is evident that this SD exhibits significantly higher protein content compared to YS2035 and IM. This characteristic stands out distinctly, further emphasizing the significance of our findings. In the initial report, the protein content of SD was significantly higher than the reported protein content (48.2%) in the reference [38].

  1. It is suggested that the genetic effects of related candidate genes should be analyzed and annotated to improve the reference value of the paper.

Response: Several studies have shown that protein plays an important role in carbohydrate and starch biosynthesis and is also associated with Golgi vesicles and vacuolar processes. In this study, several potential candidate genes were identified, including Glyma.15g108000 (the starch/carbohydrate-binding module), Glyma.18g200500 (vacuolar cation/proton exchanger 3), and Glyma.20g087600 (endoplasmic reticulum to Golgi vesicle-mediated transport), which have been discussed. Wang et al. (2021) and Kim et al. (2023) have reported Glyma.20g085700, Glyma.20g 087600, and Glyma.20g 088600 to be related to soy protein. Moreover, genes related to the vacuolar or endoplasmic reticulum to Golgi vesicle transport processes have been shown to be associated with starch and sucrose metabolism during soybean seed development. However, apart from Glyma.20g 087600, none of the other genes identified in this study have been characterized earlier.

We agree that functional characterization of these genes would improve the reference value of our work; however, the focus of the present study was to delineate new/consensus regions contributing to protein content, considering SD as the novel source. We agree with your valuable suggestion and will address this in our future study. We have indicated the same in the conclusion section of the revised manuscript as follows:

(Line398) add

“However, the genes identified in this study need further characterization and functional validation for their potential application in marker-assisted selection strategies to develop improved soybean varieties with enhanced protein content.

  1. If possible, it is recommended to supplement the results of functional identification experiments for key candidate genes.

Response: Thank you for your insightful suggestion. As mentioned in our previous response, we plan to address these points in our upcoming study. We aim to gain deeper insights into the subject and develop high-protein markers. Your feedback and suggestions are greatly appreciated as we continue our investigation. Thank you for your kind consideration and invaluable opinion.

Reviewer 2 Report

Dear Authors, the article is interesting and well written, being an interest topic for the agriculture.

This research identified the quantitative trait locus (QTL) specifically linked to seed protein content using two populations of RILs derived from SD over three years. The lack of thorough validation in diverse genetic backgrounds and limited utilization in practical breeding programs has been a challenge.

The methods are described in sufficient detail to understand the approach used and are appropriate to the statistical tests applied.

The results and disscusion support the conclusions and are shown directly and in supplementary files according to the standards of the field.

The conclusions are a reasonable extension of the results.

I appreciate that references are very recent.

Overall I think this is a very good article, with important and a high volume of scientific information and I want to congratulate the authors.

Author Response

Special Issue Editor

Plants

Special Issue: QTL Mapping of Seed Quality Traits in Crops

Dear Editor:

We wish to re-submit the manuscript titled “QTLs and Candidate Genes for Seed Protein Content in Two Recombinant Inbred Line Populations of Soybean.” The manuscript ID is plants-2522650.

We thank you and the reviewers for your thoughtful suggestions and insights. The manuscript has benefited from these insightful suggestions. I look forward to working with you and the reviewers to move this manuscript closer to publication in the Plants; Special Issue: QTL Mapping of Seed Quality Traits in Crops.

The manuscript has been rechecked, and the necessary changes have been made following the reviewers’ suggestions. The responses to all comments have been prepared and attached herewith/given below. We hope the revisions and responses have satisfactorily addressed the concerns raised by the reviewers.

I appreciate your consideration. I look forward to hearing from you.

Sincerely,

Jeong-Hyun Seo

Upland Crop Breeding Research Division, Department of Southern Area Crop Science, National Institute of Crop Science, Rural Development Administration

Miryang, Republic of Korea, 50424

+82-55-350-1277

+82-55-353-3050

[email protected]

Response to Reviewer2 comments

Dear Authors, the article is interesting and well written, being an interest topic for the agriculture. This research identified the quantitative trait locus (QTL) specifically linked to seed protein content using two populations of RILs derived from SD over three years. The lack of thorough validation in diverse genetic backgrounds and limited utilization in practical breeding programs has been a challenge.

The methods are described in sufficient detail to understand the approach used and are appropriate to the statistical tests applied.

The results and disscusion support the conclusions and are shown directly and in supplementary files according to the standards of the field.

The conclusions are a reasonable extension of the results.

I appreciate that references are very recent.

Overall I think this is a very good article, with important and a high volume of scientific information and I want to congratulate the authors.

Response: Dear Reviewer,

We hope this letter finds you well. We are writing to express our sincere appreciation for your time and expertise in reviewing our research paper. We firmly believe that our study holds significant value in the field of soybean protein research.

The authors of the paper also recognize the importance of our findings and are grateful for your positive feedback and interest in further research. Your encouragement motivates us to delve deeper into this subject and explore additional avenues of investigation.

Currently, we are actively pursuing the development of markers related to soybean protein based on the insights gained from this study. We acknowledge that this endeavor requires careful guidance and critical input from experts like yourself. Once again, we extend our heartfelt gratitude for your time and effort in reviewing our manuscript.

Thank you very much.

Reviewer 3 Report

The manuscript "QTLs and Candidate Genes for Seed Protein Content in Two Recombinant Inbred Line Populations of Soybean" reported the research of identifying QTL conferring seed protein content in three soybean lines. This is a typical QTL analysis research and could be useful for breeding that using marker-assisted selection for improving seed protein concentration. However, the manuscript is very poorly prepared. It should be reconsidered after major revisions. Following are my comments and suggestions:

Introduction:

1.       First paragraph of the Introduction has too much content not related to this research; they should be removed.

2.        The research background in the introduction is a big messy. Do you mean no protein QTL were detect before the reference genome was obtained? It should follow a logic way such as QTL from traditional linkage analysis, QTL from high saturated SNP linkage maps, GWAS, possible mechanisms of genetic control. The part of talking about materials of US, China and Korea can be stand alone and combined with Danbaekkong and SD to talk about the objectives of this research.

Results:

1.       It’s very boring to simply list all protein contents of RILs in each year. you already have tables and histograms to show the results, why need to list each one again in the text? Tables 1 and 3 are wrongly put in discussion part. I wonder if it’s necessary to list Table 1 since Figure 1 is very clear to show the results.

2.       Portion of linkage maps can add contents of comparing genetic difference among YS, SD and IM, which will be better for understanding the common and different genomic regions of the three lines, and thus will be helpful for understanding QTLs from different regions.

3.       Results of QTL should be parallelly aligned between two population to compare since the two RIL populations have a common parent, and the same genotyping method was used. The name of the QTL should be named according to parents rather than the population. For example, if Q15 is contributed by SD, simply name it as QSD15 no matter in which RIL, similarly, QSY11 means the QTL is contributed by SY, in this way, no need to use “+” or “-“ before additive effects to indicate the origin of the QTL.

4.       SNP Variation Analysis and Variant Annotation doesn’t make sense. The reference genome is not from any of the three parental lines, does the gene in the QTL region in reference genome will be the candidate gene? Are you sure the genes affecting protein content in any of the three parental lines can be found in the reference genome? If so, why need to do QTL? If not, how can you call them as candidate genes?

Materials and Methods

1.       How many replications for each RIL in each year?

2.       How many times you tested protein content for each RIL? How did you manage the data for analysis?

English is Okay but may need more editing.

Author Response

Special Issue Editor

Plants

Special Issue: QTL Mapping of Seed Quality Traits in Crops

Dear Editor:

We wish to re-submit the manuscript titled “QTLs and Candidate Genes for Seed Protein Content in Two Recombinant Inbred Line Populations of Soybean.” The manuscript ID is plants-2522650.

We thank you and the reviewers for your thoughtful suggestions and insights. The manuscript has benefited from these insightful suggestions. I look forward to working with you and the reviewers to move this manuscript closer to publication in the Plants; Special Issue: QTL Mapping of Seed Quality Traits in Crops.

The manuscript has been rechecked, and the necessary changes have been made following the reviewers’ suggestions. The responses to all comments have been prepared and attached herewith/given below. We hope the revisions and responses have satisfactorily addressed the concerns raised by the reviewers.

I appreciate your consideration. I look forward to hearing from you.

Sincerely,

Jeong-Hyun Seo

Upland Crop Breeding Research Division, Department of Southern Area Crop Science, National Institute of Crop Science, Rural Development Administration

Miryang, Republic of Korea, 50424

+82-55-350-1277

+82-55-353-3050

[email protected]

Response to Reviewer3 comments

The manuscript "QTLs and Candidate Genes for Seed Protein Content in Two Recombinant Inbred Line Populations of Soybean" reported the research of identifying QTL conferring seed protein content in three soybean lines. This is a typical QTL analysis research and could be useful for breeding that using marker-assisted selection for improving seed protein concentration. However, the manuscript is very poorly prepared. It should be reconsidered after major revisions. Following are my comments and suggestions:

Response: Thank you for providing valuable feedback on our paper. We apologize for any inconvenience caused by the initial draft's layout. We truly appreciate your insights, which have helped us improve our research quality. We have addressed all your concerns to ensure our work meets a high standard. Once again, thank you for your contribution.

Introduction:

  1. First paragraph of the Introduction has too much content not related to this research; they should be removed.

Response: Thank you for highlighting this. We have revised the first paragraph to focus on the importance of soy protein. We believe the change will be found satisfactory. Please see the revisions on lines.

(Line30) add

Soybean (Glycine max [L.] Merr.) is an important legume crop globally known for its high-quality protein and oil content [1–3]. Asian countries, like Korea, Japan, China, and Indonesia, have a strong cultural tradition of consuming soy-based products. Recently, the consumption of traditional soy-based products has surged globally, dominating the glob-al protein market [4–6]. This substantial growth is attributed to the changing dietary preferences and the shifting behavior of consumers towards more sustainable and environ-mentally friendly food choices [7–10].

Currently, soybean protein has gained increased research interest because of its significance. Many researchers aim to explore the genetic aspects of protein traits in soybean through quantitative trait loci (QTL) and genome-wide association studies (GWAS) studies [2,4].

  1. The research background in the introduction is a big messy. Do you mean no protein QTL were detect before the reference genome was obtained? It should follow a logic way such as QTL from traditional linkage analysis, QTL from high saturated SNP linkage maps, GWAS, possible mechanisms of genetic control.

Response: Dear reviewer, we apologize for the inconvenience. However, in this study, we aimed to identify novel QTLs resourced from SD, a high-protein-containing line, instead of using ‘Danbaekkong,’ which has been used widely for most QTL studies. Accordingly, we have organized the contents in the following order.

  1. Importance of soy protein
  2. QTL studies—we have included all points you have highlighted; however, we did not elaborate on them in detail to avoid diversion from our main objective.
  3. Results of major QTLs and candidate genes
  4. GWAS using different accessions
  5. The rationale for using SD
  6. Objectives

We believe the reorganized introduction is clear and appropriately represents the present study's rationale.

Results:

  1. It’s very boring to simply list all protein contents of RILs in each year. you already have tables and histograms to show the results, why need to list each one again in the text? Tables 1 and 3 are wrongly put in discussion part. I wonder if it’s necessary to list Table 1 since Figure 1 is very clear to show the results.

Response: Thank you for highlighting this. We have deleted the data from the text to avoid redundancy and moved Table 1 as a supplementary Table 1. We agree that the data in Table 1 and Figure 1 are overlapping. Nevertheless, they are also different as Table 1 shows the Heritability of the population and the results of Student’s t-test to highlight the significant difference between the parental lines. At the same time, Figure 1 show the normal distribution of protein components in the populations. Therefore, we think both are necessary to accurately show the protein traits in this paper. However, we have moved Table 1 to a supplementary file to avoid redundancy.

  1. Portion of linkage maps can add contents of comparing genetic difference among YS, SD and IM, which will be better for understanding the common and different genomic regions of the three lines, and thus will be helpful for understanding QTLs from different regions.

Response: Despite using the SD, the linkage map displayed differences in QTL regions and sizes between the YS and SI populations. The genetic differences between the two parental lines (YS2035 and IM) contributed to the dissimilarities observed in the linkage map of the YS and SI populations.

(Line133 ) add

Despite using SD as the common parental line, the differences in the genomic length coverage by the two linkage maps could be attributed to the genetic differences between the other two parental lines (YS2035 and IM).

(Line142 ) add

The differences in detecting different QTLs in the two mapping populations could be attributed to the genetic differences between YS2035 and IM.

(Line156 ) add

A total of 221 genes were found in the five QTL regions

  1. Results of QTL should be parallelly aligned between two population to compare since the two RIL populations have a common parent, and the same genotyping method was used. The name of the QTL should be named according to parents rather than the population. For example, if Q15 is contributed by SD, simply name it as QSD15 no matter in which RIL, similarly, QSY11 means the QTL is contributed by SY, in this way, no need to use “+” or “-“ before additive effects to indicate the origin of the QTL.

Response: Based on your valuable advice, we have decided to remove the symbol and use the group name as it is. To differentiate between the same segments and chromosomes, we have designated them as -1 and -2, respectively. Based on your advice, we fixed them to qPSD15-1, qPSD18-1, qPSD20-1 etc.

  1. SNP Variation Analysis and Variant Annotation doesn’t make sense. The reference genome is not from any of the three parental lines, does the gene in the QTL region in reference genome will be the candidate gene? Are you sure the genes affecting protein content in any of the three parental lines can be found in the reference genome? If so, why need to do QTL? If not, how can you call them as candidate genes?

Response: Thank you for highlighting this. Yes, the reference map was not from any of the three parental lines. Nevertheless, we performed NGS of the three parental lines and performed a comparative analysis to identify the genes in the QTL region. The method involved utilizing the workbench provided by NCBI to identify variations within each QTL region. In addition, the sequencing method for the three parents is included in 4.5. Prediction of Novel Candidate QTL and Genes as follow: We performed whole genome sequencing of SD, YS2035, and IM using the Illumina Hiseq X sequencing platform (Illumina, San Diego, CA, USA). Reads were mapped using Bowtie 2 (v2.2.4), and variants were called with freebayes (v1.3.4). After verifying tri-parent SNPs selection based on the soybean reference genome, only genes that exhibited differences from SD were screened for SNPs in YS2035 and IM. Please see lines 375.

Response: Based on the functional annotation of the identified genes that showed polymorphism between the parental lines, we speculated the key role of those genes and identified potential genes that could be associated with protein content based on previous studies. Nevertheless, the gene need to be functionally characterized to ensure their applicability in MAS. There, we think SNP variation analysis and variant annotation are useful to narrow down the number of genes to identify the potential candidate genes.

(Line209) add

“Next, we annotated the polymorphic SNPs in the genes mapped to qPSD15-1, qPSD18-1, and qPSD20-1QTLs using whole genome sequencing of SD, YS2035, and IM. After verifying tri-parent SNPs selection based on the soybean reference genome, only genes that exhibited differences from SD were screened for SNPs in YS2035 and IM. In total, 31 genes were selected among 221 genes mapped to the intervals of the five QTLs.

Response: For a comparative analysis and identifying the genes in the QTL regions, QTL analysis was essential as we lack a reference physical map of the parental lines. Regarding the concern ‘candidate genes’, we say them ‘potential candidate genes’, which need further characterization and validation to be declared as ‘candidate genes’. We apologize, as our expression in the text could have failed to explain these points. Accordingly, we have revised the related contents as follows to ensure clarity:

(Line221) add

The annotation of these 31 genes identified their association with several biological processes, including starch biosynthetic, carbohydrate metabolic, protein phosphorylation, phosphorus metabolism, and endoplasmic reticulum to Golgi vesicle-mediated transport.

We might focus on the genes that are found to be related to the mechanism of starch and protein accumulation during seed development in soybean for further study.

Materials and Methods

  1. How many replications for each RIL in each year?

Response: We grew the parental line and mapping populations once a year and used them for experiments. The results of the experiments are depicted using the data from three consecutive years.

  1. How many times you tested protein content for each RIL? How did you manage the data for analysis?

Response: The protein analysis of each RIL individual was repeated three replications, and the results in Table 1 were statistically analyzed using R and the student’s t-test was performed. We grew the parental lines and mapping populations once a year and used them for experiments. The results of the experiments are depicted using the data from three consecutive years.

(Line348) add

The protein content was measured using fifteen milligrams of seed powder in both

And add reference [59].

(Line351) add

The protein analysis of each RIL individual was analyzed three replications per year.

Round 2

Reviewer 1 Report

  • This revised manuscript does not clearly respond to the questions raised by the reviewers.

Author Response

Dear Reviewer 1

We sincerely apologize if there were any aspects of our previous submission that did not meet your expectations or if any feedback provided was not fully addressed. We have carefully considered your comments and suggestions and have made the necessary revisions accordingly. We are now resubmitting the manuscript for your advice and consideration.

In this revised version, we have taken into account your valuable feedback and have strived to improve the clarity and overall quality of the manuscript. We hope that the changes made align better with the requirements and standards of the journal.

In this manuscript, our objective extends beyond publishing our findings; we also aim to develop marker-assisted selections, such as KASP markers, utilizing the candidate genes identified in further study. However, due to the current limitation of information on the candidate genes, we have included a statement in the abstract and discussion to highlight the need for further investigation. Please check below add lines in abstract and discussion in red words.

I kindly request your understanding regarding the current research status, where references to candidate genes may still be ambiguous. Moreover, I would like to provide an evaluation of the limitations presented in this paper, particularly concerning its potential impact on the development of gene research on protein components.

First and foremost, it is important to acknowledge that the field of gene research on protein components is continually evolving, and as such, the identification of candidate genes remains a complex and challenging task.

I sincerely appreciate your guidance and valuable insights, and I am eagerly looking forward to your positive response.

Sincerely,

Jeong-Hyun Seo

Upland Crop Breeding Research Division, Department of Southern Area Crop Science, National Institute of Crop Science, Rural Development Administration

Miryang, Republic of Korea, 50424

+82-55-350-1277

+82-55-353-3050

[email protected]

Comments and Suggestions for Authors

This revised manuscript does not clearly respond to the questions raised by the reviewers.

Response:

I apologize if my initial response was not up to your expectations. I understand the significance of the revisions and additional details made to enhance the manuscript. I have thoroughly reviewed the modified content, including additional descriptions for Table 1, modifications to the materials and methods, and the inclusion of candidate genes.

Regarding the candidate gene, I have noticed the added information highlighted in red words, which clarifies its relevance to the study. This inclusion helps to address any potential ambiguity and strengthens the paper's findings in gene research on protein content. Amid a lack of references to the candidate genes we uncovered, I have added the following sentence to the candidate key gene.

(Line21) add in abstract

Among candidate genes, Glyma.15g108000, Glyma.18g200500, and Glyma.20g087600 are potentially involved in regulating protein accumulation during seed development, particularly in the vacuole and Golgi vesicle processes, as well as starch and carbohydrate metabolism.

p.13 (Line36) in discussion

Among the potential candidate genes identified, Glyma.15g108000 (starch/carbohydrate-binding module), Glyma.18g200500 (vacuolar cation/proton ex-changer 3), and Glyma.20g087600 (endoplasmic reticulum to Golgi vesicle-mediated transport), associated with vacuole-related or endoplasmic reticulum to Golgi vesicle processes, and starch and carbohydrates during seed development, are likely to regulate protein accumulation. Furthermore, the mechanism underlying carbon and nitro-gen partitioning between proteins and carbohydrates during seed maturation, which ultimately affects protein accumulation, remains unclear [55].

Regarding data interpretation and errors, the entire paper has been thoroughly corrected and supplemented in red. We have carefully addressed all the issues and made necessary revisions to enhance the quality of the manuscript.

Reviewer 3 Report

Thanks for addressing my comments. the manuscript looks ok now, but minor revision is still needed. For example, in Lines 112-113, the authors claim both mapping populations showed a normal distribution. However, the distribution histogram in Figure 1 clearly showed the transgressive distribution in both populations but more obvious in SI. 

Thanks for addressing my comments. the manuscript looks ok now, but minor revision is still needed. For example, in Lines 112-113, the authors claim both mapping populations showed a normal distribution. However, the distribution histogram in Figure 1 clearly showed the transgressive distribution in both populations but more obvious in SI. 

Author Response

Dear Reviewer 3

I truly appreciate your meticulous review and your commitment to ensuring the rigor of the research. Your expertise has undoubtedly contributed to the refinement of my study.

I have enclosed the updated manuscript, with the revisions highlighted for your convenience. I earnestly hope that these modifications meet your esteemed expectations and adhere to the journal's high standards.

I extend my heartfelt gratitude for your invaluable feedback, which has significantly contributed to enhancing the quality of my research. Your efforts and insights are genuinely appreciated.

Sincerely,

Jeong-Hyun Seo

Upland Crop Breeding Research Division, Department of Southern Area Crop Science, National Institute of Crop Science, Rural Development Administration

Miryang, Republic of Korea, 50424

+82-55-350-1277

+82-55-353-3050

[email protected]

Comments and Suggestions for Authors

Thanks for addressing my comments. the manuscript looks ok now, but minor revision is still needed.

Response: As you advised, I made minor corrections and left the letters in red. I hope it will help you review again.

For example, in Lines 112-113, the authors claim both mapping populations showed a normal distribution. However, the distribution histogram in Figure 1 clearly showed the transgressive distribution in both populations but more obvious in SI. 

Response:

It is seen that RILs with higher protein content than SD. As you advised, I can see that there are a number of individuals showing transgressive inheritance, especially in the SI population in 2021 and 2022. Therefore, it was revised to the following sentence.

(Line113) add

The protein content (%) in both populations showed a normal distribution and slightly transgressive inheritance. However, it conformed to a transgressive inheritance in the SI population, especially in 2021 and 2022 (Figure 1).
